# Extended-spectrum β-lactamase and carbapenemase-producing *Enterobacterales* among adult patients and their family members at Tikur Anbessa Specialized Hospital, Addis Ababa, Ethiopia

Dessie Abera[1,2]*, Adane Mihret[2,3], Surafel Fentaw[4], Eyob Beyene[5],
Abel Abera Negash[2,3], Woldaregay Erku Abegaz[2]*

1 Department of Medical Laboratory Sciences, College of Health Sciences, Addis Ababa University, Addis Ababa, Ethiopia, 2 Department of Microbiology, Immunology and Parasitology, College of Health Sciences, Addis Ababa University, Addis Ababa, Ethiopia, 3 Armauer Hansen Research Institute, Addis Ababa, Ethiopia, 4 Ethiopian Public Health Institute, Addis Ababa, Ethiopia, 5 Department of Internal Medicine, School of Medicine, College of Health Sciences, Addis Ababa University, Addis Ababa, Ethiopia

* dessabera@gmail.com (DA); woldearegay.erku@aau.edu.et (WEA)

## Abstract

### Background

Extended-spectrum β-Lactamase and Carbapenemase-producing *Enterobacterales* cause severe infections and currently, they are spreading beyond hospitals and becoming a serious global health concern. They often colonize the gut silently, facilitating the transmission of resistant bacteria between patients and family members.

### Objective

We sought investigate the prevalence and molecular characteristics of Extended-spectrum β-Lactamases-producing *Enterobacterales* (ESBL-PE), Carbapenem-resistant *Enterobacterales* (CRE), and factors associated there in among admitted adult patients and their family members at Tikur Anbessa Specialized Hospital, Addis Ababa, Ethiopia.

### Methods

A case-control study was conducted among 100 patients and their respective 100 family members from February 2023 to October 2023. Stool specimens were collected and processed using standard microbiological techniques. Antimicrobial susceptibility testing and ESBL production were determined using VITEK 2 system. Carbapenemase production was tested using modified carbapenem Inactivation method, and detection of resistance genes was performed by PCR.

**Data availability statement:** All relevant data are within the manuscript and its supporting information files.

**Funding:** The author(s) received no specific funding for this work.

**Competing interests:** The authors have declared that no competing interests exist.

## Result

Intestinal colonization with ESBL-PE was higher in patients (39.0%) than their respective family members (24.0%) (P = 0.028). Among patients, ESBL production was common in *E. coli*, 40.3% and *K. pneumoniae,* 34.7% than their family members 24.2% and 22.2%, respectively. Of the ESBL-PE isolates, 84.6% from patients and 100% from family members carried at least one ESBL encoding gene, with $bla_{CTX-M}$ being the predominant. Colonization with CRE and Carbapenemase-Producing Carbapenem-Resistant *Enterobacterales* was found to be 19.0% and 10.0%, respectively. These were identified only among patients, with $bla_{NDM}$ and $bla_{OXA-48}$ are the most prevalent genes. Older age (>53 years) (P = 0.02) and previous ICU admission (P < 0.001) showed significant association with ESBL-PE colonization.

## Conclusion

ESBL-PE colonization was more prevalent in patients compared to their family members, with $bla_{CTX-M}$ identified as the most common gene. Exclusive detection of carbapenemase genes among patients, and the association of previous ICU admission with ESBL-PE colonization, highlights the need for targeted screening and strengthened infection prevention.

## Introduction

Antimicrobial resistance, specifically to third generation cephalosporin and carbapenems resistance, poses a significant global public health challenge [1]. Extended-spectrum β-Lactamase producing *E. coli* and *K. pneumoniae* are two of the most common resistant bacteria, both of which capable of colonizing the gastrointestinal tract, that may serve as a reservoir for transmission of resistance genes to other gastrointestinal resident bacteria [2]. Moreover, these bacteria are often implicated in community and hospital acquired-infections such as urinary tract infection, sepsis and pneumonia in hospitalized patients [3]. Extended-spectrum β-Lactamases (ESBLs) and Carbapenemase-Producing Carbapenem-Resistant *Enterobacterales* (CP-CRE) are plasmid-encoded enzymes that develop resistance to broad-spectrum beta-lactam antibiotics such as penicillin, cephalosporins and carbapenems [4]. The resistance genes encoding these enzymes can spread from one bacteria to another bacteria through horizontal gene transfer (HGT) [5]. The most clinically important ESBLs genes include $bla_{TEM}$, $bla_{SHV,}$ and $bla_{CTX-M}$, of which $bla_{CTX-M}$ being widely distributed in community and hospital settings [6].

Moreover, the emergence of CRE has become a critical public health challenge due to inadequate treatment options [7]. Carbapenems are often taken as the last resort antibiotics for severe infections caused by multi-drug resistant (MDR) pathogens. Carbapenemase-Producing Carbapenem-Resistant *Enterobacterales* have a potential to specifically hydrolyze carbapenems and become problematic in healthcare facilities [8]. The most common carbapenemase encoding genes are

*K. pneumoniae* carbapenemase producing (KPC), New Delhi metallo-β-Lactamase (NDM), Oxacillinases (OXA-48 like), Verona-encoded integron metalo-β-Lactamase (VIM) and Imipenemase (IMP) [9].

Prolonged antibiotic use, invasive medical procedures, intensive care unit (ICU) admission, chronic diseases, and longer hospital stays have been associated with ESBL-PE colonization [10]. Fecal carriage of ESBL producing *E. coli* and *K. pneumoniae* are frequently documented in the hospital environments worldwide [11]. In Chad, the prevalence of ESBL-PE colonization has been reported with the prevalence of 51% among hospitalized patients, and 38% among healthy individuals [12]. In Ethiopia, studies have shown ESBL-PE fecal carriage ranging from 47.3% to 52% among admitted patients [13,14], while the fecal carriage rate of CP-CRE were reported at the rate of 0.75% [13], 5.5% [15], and 7% [16]. Recently, community level fecal colonization with ESBL-PE and CRE were reported in Ethiopia, with the prevalence of 31.4% and 0.8%, respectively [17].

The spread of ESBL-PE and CRE in the community has become significant public health concern, mainly due to asymptomatic gut colonization by these bacteria [18]. Furthermore, patients colonized with ESBL-PE and CRE can serve as a source of transmission to family members, other patients, and healthcare workers. Similarly, family members may act as a silent carrier, facilitating transmission to patients, or to the hospital community [19,20]. While the majority of previous studies conducted in Ethiopia focused on fecal carriage of ESBL-PE and CRE either the community members, or hospitalized patients independently. Data on intestinal colonization of EBL-PE and CRE in both patients and their close contacts at the time of admission in Ethiopia is scarce. Therefore, we aimed to determine the intestinal colonization of ESBL-PE, CRE, CP-CRE, and factors associated there in among adult patients and their respective family members at admission in Tikur Anbessa Specialized Hospital, Addis Ababa, Ethiopia.

## Methods and materials

### Study design, setting, and study population

Hospital based case-control study was conducted from February 2023 to October 2023 at Tikur Anbessa Specialized Hospital (TASH), the largest specialized teaching hospital in Ethiopia with over 800 beds and serving more than 500,000 patients annually. All patients admitted at internal medical ward and their family members were the study population. Patients aged ≥ 18 years old who consented were included, as well as family members aged ≥ 18 years old and living in the same house hold. Patients who were critically ill or lack family members, and family members who were on antibiotic use and had history of admission within the last 3 months, were excluded.

### Sample size calculation and sampling technique

The sample size was calculated using case-control calculation formula, $n = (r+1) \times p(1-p) \times (Z\alpha/2 + Z\beta)^2/r (P1-P2)^2$, by considering the previous prevalence of ESBL-PE colonization among hospitalized patients (P1 = 42%) and healthy volunteers (P2 = 22%) in Burkina Faso. The value of $Z\alpha/2$ is 1.96 at 95% confidence level, and $Z\beta$ is set at the critical value of 0.84 with a margin of 5%. By including the 20% non-response rate, the final sample size was determined to be 85 for each participant, which gave a total of 204 sample size.

P1 = Prevalence of ESBL colonization among hospitalized patients

P2 = Prevalence of EBL colonization among healthy volunteers

P = Average prevalence (P1 + P2)/2

r = Ratio of control to cases (1 for equal number of control and cases)

P1-P2 = Effect size or different in proportion expected based on previous study

A systematic random sampling technique was used to select study participants aged 18 years and above who were admitted to internal medicine wards. We used the list of all patients who were admitted to medical wards over the past six months for the purpose of using it as our sampling frame. The total number (N) of admitted patients in the past six months was 624, and the minimum calculated sample size (n) was **204**. Therefore, K = N/n: 624/204 = 3. A random start was chosen from the first three admissions, and subsequently every third patient was enrolled until the desired sample size was achieved. However, four eligible study participants (both patients and family members) declined to participate. Therefore, a total of 200 stool samples were collected.

## Data collection

Prior to sample collection, informed consent was taken from each study participant. Socio-demographic characteristics, and associated factors that can potentially be linked to ESBL, and/or CRE colonization such as underline health conditions, duration of hospitalization, number of beds in a single room in a ward, presence of invasive medical devices, previous antibiotic exposures, previous ICU admission history and travel history were assessed using a pre-tested questionnaire.

After obtaining consent from each participant, a questionnaire was completed through a data collector-administered interview and each question was translated from English to Amharic. We explained briefly for each participant how to collect approximately 2gram fresh fecal specimen. Sterile, dry screw-top containers were used for fecal specimen collection. Fresh fecal specimens from 100 patients and 100 attendees were collected. Then a small portion of fecal samples was put into a Carry-Blair transport medium and transported to Ethiopian Public Health Institute (EPHI) within 4 hours of specimen collection at a temperature of 2–4°C.

## Bacterial isolation and identification

All fecal samples were inoculated on MacConkey agar and incubated at 35–37°C for 24 hours at aerobic condition. Bacterial growth was characterized based on their colony morphology such as size, texture, and pigmentation. Then a single pure colony was sub-cultured on tryptic soya agar (TSA), and the identity of the target bacteria was confirmed by biochemical tests such as indole production, urease, oxidase, citrate, lysine, mannitol, and triple sugar iron (TSI) [21]. In addition, bacterial identification was performed by VITEK 2 Compact system (bioMe'rieux, France), using gram-negative bacteria identification cards (GN-ID). The main principle of the VITEK 2 system is colorimetry technology that measures biochemical reactions contained in a variety of microbe identification cards. Around 3–5 bacterial colony suspension was prepared by using 3.0 ml of sterile saline (0.45%) into clear plastic (polystyrene) test tubes. The density of suspension was checked for 0.50 McFarland standard using a turbidity meter. Minimum inhibitory concentrations (MICs) were determined using the VITEK 2 compact system, and results were interpreted as sensitive, intermediate and resistant [22]. Bacterial isolates were stored in tryptic soya broth (TSB) with 20% of glycerin at-80°C for further analysis.

## Antimicrobial susceptibility testing (AST)

Antimicrobial susceptibility tests were performed using VITEKE 2 AST-GN cards. The following antimicrobials were included: ampicillin (AMP), amoxicillin-clavulanic acid (AMC), ampicillin/sulbactam (AMS), cefazolin (CXF), cefuroxime (CXM), cefuroxime-axetil (CAE), ceftazidime (CAZ), ceftriaxone (CRO), cefepime (CFP), ertapenem (ERT), imipenem (IPM), gentamicin (GEN), ciprofloxacin (CIP), tobramycin (TXM), levofloxacin (LVF), tetracycline (TXC), nitrofurantoin (NIF), trimethoprim-sulfamethoxazole (SXT). The density of 3–5 colony suspension was checked for 0.50 McFarland standard using a turbidity meter and the suspension was transferred to clean and sterile test tube for AST and the results were interpreted according to CLSI protocol [23].

 

## Phenotypic detection of ESBL production

ESBL production was identified using VITEK 2 compact system in accordance with manufacturer instructions. AST-GN86 cards contain cefotaxime (0.5 μg/ml), ceftazidime (0.5 μg/ml), cefepime (1 μg/ml), and β-lactam inhibitors such as cefotaxime/Clavulanic Acid (0.5/4 μg/ml), ceftazidime/Clavulanic Acid ((0.5/4 μg/ml) and cefepime/Clavulanic Acid ((1/10 μg/ml) were used to detect ESBL production [22].

## Phenotypic detection of carbapenemase production

All isolates resistant to carbapenem (imipenem and/or ertapenem) were tested for carbapenemase production by modified carbapenem Inactivation method (mCIM). About 1–4 loops full of CRE isolates were suspended in 2 mL of tryptic soy broth and 10 μg meropenem disk was submerged in each tube and incubated at ambient temperature for 4 hours ± 15 minutes. Fresh carbapenem susceptible ATCC25922 *E. coli* was suspended with sterile saline and checked for an equivalent 0.50 McFarland with densitometer and spread onto Muller-Hinton agar. Then, meropenem was removed from each tube and placed on Muller-Hinton agar and incubated at 35°C ± 2°C for 18–24 hours and result interpretation was made based on CLSI 2023 guideline [23]. Measure of inhibition zone with diameter of 6–15 mm or pinpoint colonies in 16–18 mm were determined as positive for carbapenemase enzyme production; and a zone of inhibition ≥19 mm was considered to be negative for carbapenemase enzyme production.

## Molecular detection of β-lactamase genes

**Bacterial DNA extraction.** Heat lysis technique was used to extract the DNA of bacterial isolates [24]. Bacterial isolates were inoculated on Tryptone Soy Agar (TSA) and incubated aerobically at 37°C for 24 hours. A loop full (1 μL loop) of fresh colonies were suspended in 100 μL Tris-acetate-EDTA (TAE) buffer and heated at 100°C for 10 minutes. Then, maintained at ambient temperature for 5 minutes to cool down and centrifuged at 14,000g for 5 minutes at 4°C. About 50 μL of the supernatant was transferred into nuclease free Eppendorf tube and stored at −20°C until analysis for short time and at −80°C for long period.

**PCR amplification.** Conventional polymerase chain reaction (PCR) was used to detect β-Lactamase encoding genes at Armauer Hansen Research Institute (AHRI), Addis Ababa, Ethiopia. The PCR protocol was used from the previous studies [25,26]. β-Lactamase genes were detected by multiplexing; group-1 multiplex ($bla_{TEM}$, $bla_{CTX-M}$, and $bla_{OXA-23-like genes}$) and group-2 multiplex ($bla_{GES}$, $bla_{VEB}$, $bla_{AmpC}$). Similarly, carbapenemase genes were multiplexed; carba group-1 multiplex ($bla_{KPC}$ $bla_{NDM}$, $bla_{OXA-48}$, $bla_{BIC}$), carba group-2 multiplex ($bla_{VIM}$, $bla_{SPM}$, $bla_{IMP}$), and carba group-3 multiplex ($bla_{AIM}$, $bla_{GIM}$, $bla_{SIM}$, $bla_{DIM}$), and uniplex analysis was performed to detect $bla_{SHV}$. The primers were grouped based on similar annealing temperature requirements and primer compatibility (efficient co-amplification under a single PCR condition). To prevent band overlapping and to facilitate clear interpretation of results, primers with varying amplicon sizes were grouped together.

Amplification reactions were performed in a final volume of 25 μL containing 12.5 μL 2x Tag plus PCR Master Mix (Thermo Scientific, Lithuania, EU), 1 μL of DNA primers (0.5 μL forward plus 0.5 μL reverse primers), 5.5 μL of nuclease free water; and 5 μL of template DNA preparations were added to the reaction mixture. Reactions were performed in a DNA thermal cycler (T100 BIO-RAD, USA) under the following conditions: denaturation at 95°C for 5 minutes followed by 40 cycles at 95°C for 20 seconds, 61°C for 30 seconds, and 72°C for 1 minute with a final extension of 72°C for 7 minutes [27]. After PCR amplification, 2.5 μL of each reaction was separated by gel electrophoresis in 1.5% agarose gel for 50 minutes at 120 V in 0.5 × TAE buffer. DNA was stained with ethidium bromide (1.5 μg/mL), and the bands were detected using a UV transilluminator (Cleaver Scientific Ltd, Rugby, UK.). β-Lactamase and CP-CRE encoding genes were amplified using specific primers as indicted in (S1Table) [25], and (S2 Table) [26], respectively.

## Quality control

Prior to use, the expiration dates of the media, reagents, and antibiotic disks were checked. Additionally, visual checks were conducted to ensure there were no cracks, bubbles, and contaminants of the culture media. ATCC 25922 *E. coli* and ATCC 700603 *K. pneumoniae* were used as a negative and positive control strains, respectively, as per CLSI guideline [23]. Positive and negative quality controls were performed using *K. pneumoniae* ATCC BAA-1705 and *K. pneumoniae* ATCC BAA-1706 strains, respectively, to check the quality of antibiotic disks. Known *E. coli* strains carrying $bla_{TEM}$, $bla_{SHV}$, $bla_{CTX-M}$, and *Klebsiella pneumoniae* isolates bearing $bla_{KPC}$, $bla_{NDM,}$ and $bla_{OXA-48}$ were used for antibiotic resistance gene detection.

**Operational definition. CRE**: *Enterobacterales* that are non-susceptible to the antibiotic class of carbapenems.

**CP-CRE**: *Enterobacterales* that can produce carbapenemase enzyme.

**Colonization rate or carriage rate:** The presence of ESBL-PE or CP-CRE in the gut.

**ESBL-PE:** Extended-spectrum β-Lactamase producing *Enterobacterales.*

**MDR:** Multidrug resistance is defined as non-susceptible to ≥1 agent in >3 antimicrobial categories [28].

## Statistical analysis

All data were entered in Microsoft Excel 2016 and exported to SPSS version 25 (IBM corporate, USA). The proportion of bacterial isolates and the associated factors of ESBL-PE colonization were described using descriptive statistics, such as frequencies and percentages. The association between ESBL-PE colonization and associated factors was investigated using binary logistic regression. ESBL-PE colonization was also compared between patients and family members using chi-square ($X^2$) and binary logistic regression. Furthermore, all variables with a p-value of <0.25 in the bivariable analysis were included in multivariable analysis to account for possible confounding variables. The P-value less than 0.05 is considered statistically significant. The strength of the association was interpreted using an odds ratio in a 95% confidence interval. Finally, the results presented on words, graphs, and tables.

## Ethical approval

This study was reviewed and approved by the intuitional review board (IRB) of the College of Health Science, Addis Ababa University (Protocol number:072/22/DMIP). Permission letter was secured from Tikur Anbessa Specialized Hospital. Written informed consent was obtained from the study participants before sample collection. The aim of the study, the study procedures, possible risks/benefits and the right to withdraw from the study at any time was explained for the study participants, and their data were kept confidentially. This study was conducted based on the declaration of Helsinki.

## Results

### Socio-demographic characteristics of the study participants

The total number of study participants were 200 (100 patients and 100 respective family members). From the total patients, 51.0% (51/100) were males, and similar number of males were observed in family members, 52.0% (52/100). The median age of patients and family members were 36 and 32, respectively (range from 18 to 63 years old). Majority of patients, 51.0% (51/100), had 2–3 household members. About 66.0% (66/100) participants were from Addis Ababa city. Most of the patients had a higher education level, 34.0% (34/100), and 60.0% (60/100) of them had marriage (Table 1).

### Clinical characteristics of patients

The primary reason for patient's admission was acute lymphoid leukemia (ALL), accounted for 35.0% (35/100). This was followed by acute myeloid leukemia 18.0% (18/100), and liver disease 10.0% (10/100). Majority of the patients had

**Table 1. Socio-demographic characteristics of patients and their family members in Tikur Anbessa Specialized Hospital in Addis Ababa, Ethiopia from February 2023 to October 2023 (n = 200).**

| Variables | | Patients (n = 100) | Family members (n = 100) |
|---|---|---|---|
| | | Frequency (%) | Frequency (%) |
| Sex | Female | 49 (49.0) | 48 (48.0) |
| | Male | 51 (51.0) | 52 (52.0) |
| Age category | 18-29 | 31 (31.0) | 43 (43.0) |
| | 30-41 | 32 (32.0) | 31 (31.0) |
| | 42-53 | 27 (27.0) | 16 (16.0) |
| | >53 | 16 (16.0) | 10 (10.0) |
| Residence | Addis Ababa | 66 (66.0) | 66 (66.0) |
| | Out of Addis Ababa | 34 (34.0) | 34 (34.0) |
| Occupation | Government | 29 (29.0) | 32 (32.0) |
| | Private | 28 (28.0) | 29 (29.0) |
| | Merchant | 13 (13.0) | 14 (14.0) |
| | Farmer | 28 (15.2) | 20 (20.0) |
| | Student | 15 (15.0) | 5 (5.0) |
| Education | Informal education | 27 (27.0) | 31 (31.0) |
| | Primary school | 9 (9.0) | 11 (11.0) |
| | High school | 30 (30.0) | 20 (20.0) |
| | Higher education | 34 (34.0) | 38 (38.0) |
| Family number | 1 | 14 (14.0) | 11 (11.0) |
| | 2-3 | 51 (51.0) | 49 (49.0) |
| | 4-5 | 20 (20.0) | 30 (30.0) |
| | >5 | 15 (15.0) | 10 (10.0) |
| Marital status | Single | 36 (36.0) | 34 (31.0) |
| | Married | 60 (60.0) | 57 (56.0) |
| | Divorced | 4 (40.0) | 9 (9.0) |

underlining health conditions, HIV being the most common underlying condition, 16.0% (16/100), followed by cardiac problem, 12.0% (12/100) (Table 2).

**The proportion of *Enterobacterales* among patients and their family members.** A total of 107 Enterobacterales isolates were detected from 100 patients, and 106 isolates from 100 family members. Of the total 107 isolates from patients, 71.9% (77/107) and 21.4% (23/107) were *E. coli* and *K. pneumoniae*, respectively. Of these, 40.3% (31/77) of *E. coli* and 34.7% (8/23) of *K. pneumoniae* were ESBL producers. In family members, *E. coli* was the predominant bacterium, accounted for 85.8% (91/106) followed by *K. pneumoniae* at 8.5% (9/106). Among these, 24.2% (22/91) of *E. coli* and 22.2% (2/9) of *K. pneumoniae* were ESBL producing bacteria. Moreover, mixed population of *Enterobacterales* were also detected: *E. coli + K. pneumoniae,* 1.8% (2/107) in patients, and 0.9% (1/106) in family members; equal number of *E. coli + C. freundii* isolates (two each) were detected among both study groups, making the proportion of 1.8% for patients and 1.9% for family members (Fig 1).

**Prevalence of intestinal colonization with ESBL-PE among patients at admission and their family members.** The overall intestinal colonization with ESBL-PE among patients and their family members was 63.0% (63/200). Patients have a higher ESBL-PE colonization, 39.0% (39/100) than family members, 24.0% (24/100) ($x^2$ = 5.21, P = 0.028). Patients are 1.63 times more likely to be colonized with ESBL-PE than their family members (COR = 1.63, 95% CI = 1.06–2.49, P = 0.028). ESBL producing *E. coli* was dominantly found more from patient isolates, 40.3% (31/77)

**Table 2. Clinical characteristics of patients at admission in Tikur Anbessa Specialized Hospital in Addis Ababa, Ethiopia from February 2023 to October 2023 (n = 100).**

| Clinical conditions of | Patients (n = 100) |
|---|---|
| **Reason for admission** | Frequency (%) |
| Renal disease | 8 (8.0) |
| Liver disease | 10 (10.0) |
| Hemorrhagic stroke | 4 (4.0) |
| AML | 18 (18.0) |
| ALL | 35 (35.0) |
| CML | 3 (3.0) |
| NHL | 3 (3.0) |
| Colon cancer | 3 (3.0) |
| Cardiac problem | 10 (10.0) |
| GI problem | 6 (6.0) |
| **Underlining conditions** | |
| Renal disease | 11 (11.0) |
| HIV | 16 (16.0) |
| Live disease | 8 (8.0) |
| Hypertension | 6 (6.0) |
| Neurologic problem | 7 (7.0) |
| Cardiac problem | 12 (12.0) |
| Type II diabetics | 6 (6.0) |

AML= Acute myeloid leukemia, ALL= Acute lymphoid leukemia, CML= Chronic myeloid leukemia, NHL= Non-Hodgkin lymphoma.

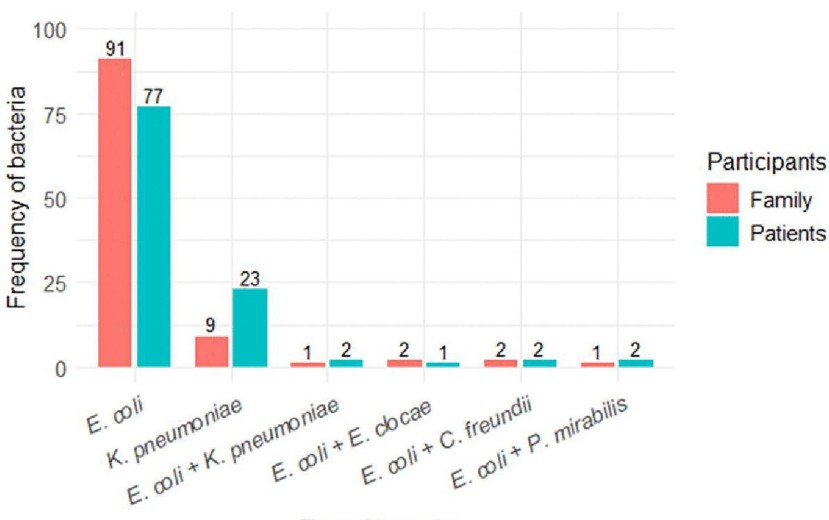

**Fig 1. Frequency of *Enterobacterales* among patients at admission and their family members.**

than isolates from family members, 28.2% (22/91), with significant association ($x^2 = 4.28$, P = 0.04), and (COR = 4.28, 95% CI = 1.09–4.10, P = 0.04). ESBL producing *K. pneumoniae* isolates were also more common in patients, 34.8% (8/23) compared to isolates from family members, 22.2% (2/9). However, because of the limited number of ESBL producing *K. pneumoniae* isolates identified in family members, additional statistical analysis was not performed (Table 3).

**Prevalence of intestinal colonization with CRE and CP-CRE among patients at admission.** The overall intestinal colonization with CRE and CP-CRE among patients at admission were 19.0% (19/100) and 10.0% (10/100), respectively. The proportion of CRE *E. coli* was 16.8% (13/77), of which 38.5% (5/13) were CP-CRE. More CRE was detected in *K. pneumoniae* with the proportion of 46.2% (6/23), among this 83.3% (5/6) were positive for CP-CRE. However, CRE was not detected in family members (Fig 2).

**Phenotypic antimicrobial resistance of ESBL-PE isolates from patients and their family members.** ESBL producing *E. coli* from patients and their attending family members were resistant to cefazolin at the rate of 100% (31/31) and 95.5% (21/22), respectively; however, the difference between the two was not statistically significant (P > 0.05). This bacterium showed high resistant to ceftriaxone, with rates of 93.5% (29/31) among patients and 90.9% (20/22) among family members. Moreover, higher resistance rate was observed in ESBL producing *E. coli* against gentamycin among patients, 61.3% (19/31) than family members, 45.5% (10/22). Similarly, ESBL producing *K. pneumoniae* was highly resistant to cefazolin, ceftazidime and ceftriaxone at the rate of 100% (8/8) among patients and 100% (2/2) in family members. However, statistical analysis was not performed due to the limited number of ESBL *K. pneumoniae* isolates. ESBL producing *E. coli* showed resistance to nitrofurantoin more in patients than in family members with statistically significant difference at the rate of 54.8% (17/31), and 27.3% (6/22) (P = 040), respectively (Table 4).

**The proportion of ESBL-PE and CP-CRE encoding genes among patients and their family members.** Among the 39 ESBL-PE isolates obtained from patients, 84.6% (33/39) carried at least one ESBL encoding gene. However, all 24 ESBL-PE isolates from family members were positive for one or more ESBL encoding genes. $bla_{CTX-M}$ was the most frequently detected gene in ESBL producing *E. coli* among patients, 74.2% (23/31) while it was 68.2% (15/22) among their family members. On the other hand, the proportions of $bla_{TEM}$ in ESBL producing *E. coli* were almost similar between patients, 38.7% (12/31), and family members, 40.9% (9/22). $bla_{SHV}$ was detected less frequently in ESBL producing *E. coli* in both patients and family members at 32.2% (10/31) and at 22.7% (5/22), respectively. Additionally, high proportion

**Table 3. Chi-square and bivariate analysis of intestinal colonization with ESBL-PE among patients and their family members at Tikur Anbessa Specialized Hospital in Addis Ababa, Ethiopia from February 2023 to October 2023 (n = 200).**

| Study group | ESBL | | | | | | | |
| --- | --- | --- | --- | --- | --- | --- | --- | --- |
| | Positive n (%) | Negative n (%) | Total | $X^2$ | P-value | COR | 95% CI | P-value |
| Patients | 39 (39.0) | 61 (61.0) | 100 | 5.21 | 0.028 | 1.63 | 1.06-2.49 | 0.028 |
| Family members | 24 (24.0) | 76 (76.0) | 100 | | | | | |
| | *E. coli* + n (%) | *E. coli* - n (%) | | | | | | |
| Patients | 31 (40.3) | 46 (59.7) | 77 | 4.28 | 0.04 | 2.11 | 1.09-4.10 | 0.04 |
| Family members | 22 (24.2) | 69 (75.8) | 91 | | | | | |
| | *K. pneumoniae* + n (%) | *K. pneumoniae* - n (%) | | | | | | |
| Patients | 8 (34.8) | 15 (65.2) | 23 | | | | | |
| Family members | 2 (22.2) | 7 (77.8) | 9 | | | | | |

*E. coli* += ESBL producing *E. coli*, *E. coli* - = non-ESBL *E. coli*.

*K. pneumoniae* += ESBL producing *K. pneumoniae*, *K. pneumoniae* - = non-ESBL *K. pneumoniae*.

$X^2$ = Chi-square test, COR = Crude odds ratio, CI = Confidence interval.

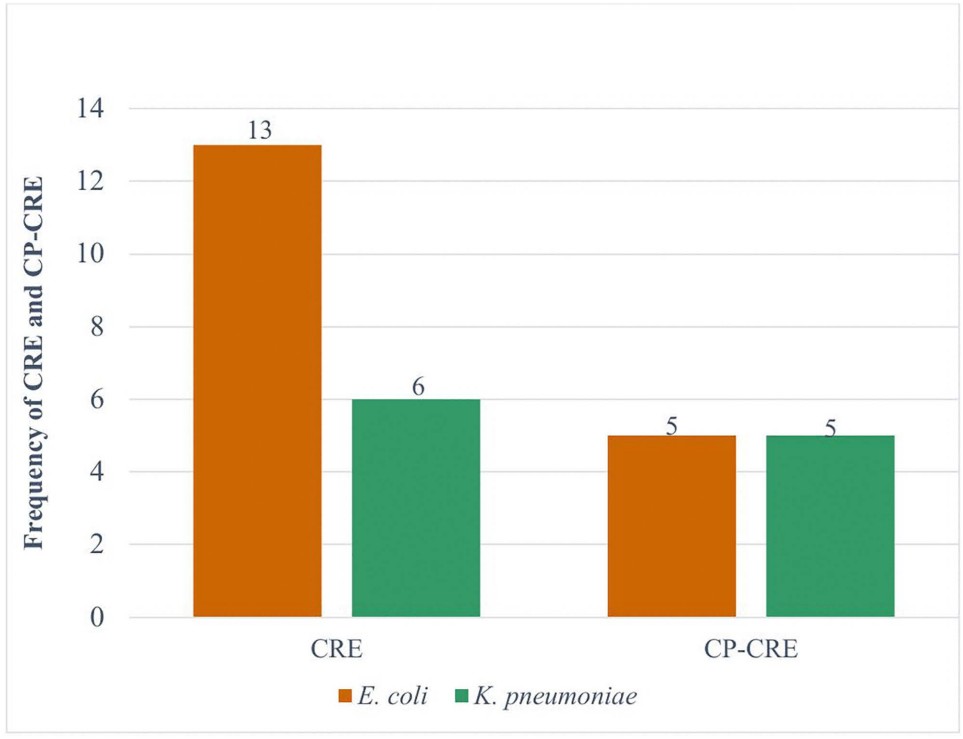

**Fig 2. Frequency of CRE and CP-CRE in *E. coli* and *K. pneumoniae* among patients at admission.**

of $bla_{CTX-M}$ was detected in ESBL producing *K. pneumoniae* among both patients and their families with the proportion of 87.5% (7/8) and 100% (2/2), respectively. Similarly, $bla_{TEM}$ in ESBL producing *K. pneumoniae* was identified at 62.5% (5/8) and 100% (2/2) among patients and family members (Table 5).

Furthermore, the combination of ESBL genes were detected from both *E. coli* and *K. pneumoniae* among both patients and their family members. For example, $bla_{TEM} + bla_{CTX-M}$, and $bla_{CTX-M} + bla_{SHV}$ was found in ESBL producing *E. coli* among 22.5% (7/31) and 18.2% (4/22) of patients and 12.9% (4/31) and 9.1% (2/22) of family members, respectively. From the total 19 CRE isolates among patients, 84.2% (16/19) were positive for at least one or more carbapenemase encoding genes. $bla_{NDM}$ and $bla_{OXA-48}$ genes in CRE *K. pneumoniae* were entirely found from patients only at the rates of 83.3% (5/6) and 50.0% (3/6), respectively. These carbapenemase genes were also commonly identified in CRE *E. coli* with the proportion of 30.7% (4/13) and 23.1% (3/13), respectively (Table 5).

Moreover, gel electrophoresis analysis shows the PCR amplification products of β-Lactamase encoding genes in *E. coli* and *K. pneumoniae* isolates as shown in (Figs 3, 4), respectively.

**Associated factors and ESBL-PE colonization among patients and their family members.** Patients with the age of >53 years had higher odds ratio to be colonized with ESBL-PE than the other age groups (AOR = 8.91, 95% CI = 1.52–52.3, P = 0.02). Similarly, patients who had a history of ICU admission in the past 6 months were more likely to be colonized by ESBL-PE (AOR = 12.4, 95% CI = 3.98–38.6, P < 0.001) (Table 6). However, ESBL-PE colonization in family members were not statistically associated with factors such as age, sex, previous antibiotic use and admission history (P > 0.05) (Table 7).

**Associated factors and CRE colonization among patients.** Patients who had chronic diseases such as liver and renal diseases were more likely to be colonized with CRE, although the difference did not reach a statistically significant level. Other potential factors were not statistically associated with CRE colonization (P > 0.05) (Table 8).

**Table 4. Antimicrobial resistance in ESBL producing *E. coli* and *K. pneumoniae* among patients on admission and their family members at Tikur Anbessa Specialized Hospital in Addis Ababa, Ethiopia from February 2023 to October 2023 (n = 200).**

| Category | Antimicrobial agents | ESBL producing *E. coli* | | X² | P-value | ESBL producing *K. pneumoniae* | |
|---|---|---|---|---|---|---|---|
| | | Patients (n = 31) | Families (n = 22) | | | Patients (n = 8) | Families (n = 2) |
| | | Resistance n (%) | Resistance n (%) | | | Resistance n (%) | Resistance n (%) |
| Aminoglycosides | GM | 19 (61.3) | 10 (45.5) | 1.26 | 0.262 | 5 (62.5) | 1 (50) |
| | TOB | 17 (54.8) | 11 (50) | 0.12 | 0.726 | 2 (25.0) | 1 (50) |
| Cephalosporines | CFZ | 31 (100) | 21 (95.5) | 1.45 | 0.228 | 8 (100) | 2 (100) |
| | CFX | 31 (100) | 22 (100) | | | 8 (100) | 2 (100) |
| | CFA | 31 (100) | 21 (95.5) | 1.45 | 0.228 | 8 (100) | 2 (100) |
| | CAZ | 29 (93.5) | 19 (86.4) | 0.77 | 0.380 | 8 (100) | 2 (100) |
| | CTX | 29 (93.5) | 20 (90.9) | 0.13 | 0.714 | 8 (100) | 2 (100) |
| | FEP | 27 (87.1) | 17 (77.2) | 0.87 | 0.352 | 6 (75.0) | 1 (50) |
| Carbapenem | ERT | 6 (19.4) | 0 | | | 4 (50) | 0 |
| | IMP | 3 (9.7) | 0 | | | 4 (50) | 0 |
| Fluoroquinolone | CRO | 18 (58.1) | 9 (40.9) | 1.51 | 0.219 | 6 (75.0) | 1 (50) |
| | LEV | 16 (51.6) | 11 (54.5) | 0.01 | 0.915 | 5 (62.5) | 1 (50) |
| Sulfonamides | TXS | 22(70.9) | 13 (59) | 0.81 | 0.367 | 6 (75.0) | 2 (100) |
| Penicillin | AMP | 30 (96.8) | 20 (90.9) | 0.86 | 0.354 | 8 (100) | 2 (100) |
| Penicillin+β-lactamase inhibitors | SAM | 11 (35.5) | 13 (59) | 0.01 | 0.883 | 6 (75.0) | 0 |
| | AMC | 14 (45.2) | 10 (45.5) | 2.84 | 0.092 | 4 (50) | 0 |
| Tetracyclines | TET | 31 (100) | 21 (95.5) | 1.45 | 0.228 | 8 (100) | 2 (100) |
| Nitrofurantoin | NT | 17 (54.8) | 6 (27.2) | 3.97 | 0.040 | 4 (100) | 2 (100) |

AMP = Ampicillin, SAM = Ampicillin Sulbactam, AMC = Amoxicillin/Clavulanic acid, CFZ = Cefazolin, CFX = Cefuroxime, CFA = CefuroximeAxetil, CAZ = , Ceftazidime, CTX = Ceftriaxone, FEP = Cefepime, ERT = Ertapenem, IMP = Imipenem, GM = Gentamycin, TOB = Tobramycin, CRO = Ciprofloxacin, LEV = Levofloxacin, TET = Tetracycline, NIT = Nitrofurantoin, TXS = Trimethoprim/Sulfamethoxazole.

**Table 5. The proportion of ESBL-PE and CRE encoding genes among patients and their family members at Tikur Anbessa Specialized Hospital in Addis Ababa, Ethiopia from February 2023 to October 2023 (n = 200).**

| Genes detected | Patients (n = 100) | | Total | Family members (n = 100) | | Total |
|---|---|---|---|---|---|---|
| | ESBL *E. coli* (n = 31) n (%) | ESBL *K. pneumoniae* (n = 8) n (%) | | ESBL *E. coli* (n = 22) n (%) | ESBL *K. pneumoniae* (n = 2) n (%) | |
| $bla_{TEM}$ | 12 (38.7) | 5 (62.5) | 17 (43.5) | 9 (40.9) | 2 (100) | 11 (45.8) |
| $bla_{CTX-M}$ | 23 (74.2) | 7 (87.5) | 30 (76.9) | 15 (68.2) | 2 (100) | 17 (70.8) |
| $bla_{SHV}$ | 10 (32.2) | 5 (62.5) | 15 (38.5) | 5 (22.7) | 1 (50.0) | 6 (25.0) |
| $bla_{TEM} + bla_{CTX-M}$ | 7 (22.5) | 5 (62.5) | 12 (30.7) | 4 (18.2) | 1 (50.0) | 5 (20.8) |
| $bla_{TEM} + bla_{SHV}$ | 2 (6.5) | 4 (50.0) | 6 (15.4) | 1 (4.5) | 1 (50.0) | 2 (8.3) |
| $bla_{CTX-M} + bla_{SHV}$ | 4 (12.9) | 2 (25.0) | 6 (15.4) | 2 (9.1) | 1 (50.0) | 3 (12.5) |
| | CRE *E. coli* (n = 13) | CRE *K. pneumoniae* (n = 6) | Total | | | |
| $bla_{NDM}$ | 4 (30.7) | 5 (83.3) | 9 (47.4) | | | |
| $bla_{KPC}$ | 1 (7.6) | 2 (33.3) | 4 (21.1) | | | |
| $bla_{OXA-48}$ | 3 (23.1) | 3 (50.0) | 7 (36.8) | | | |
| $bla_{VIM}$ | 1 (7.6) | 1 (16.7) | 2 (10.5) | | | |
| $bla_{IMP}$ | 2 (15.4) | 1 (16.7) | 3 (15.8) | | | |

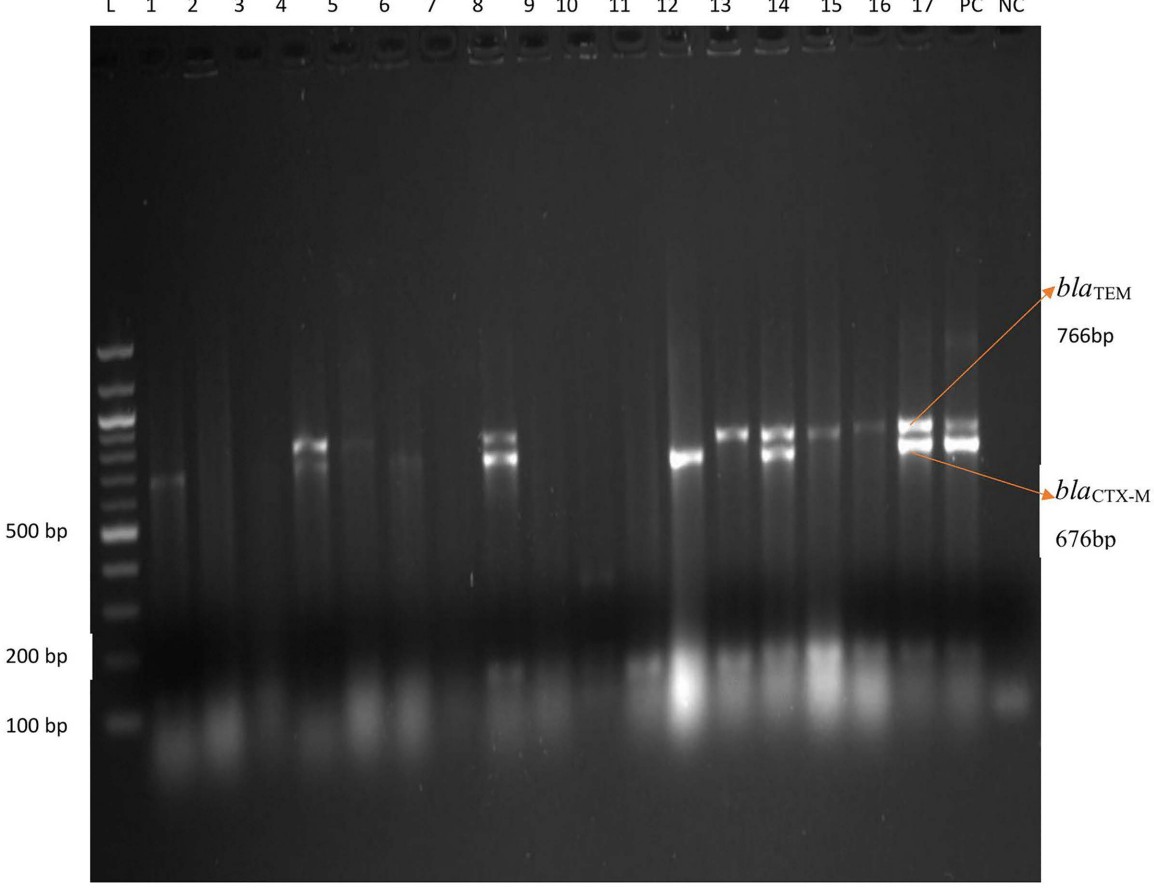

**Fig 3. Gel electrophoresis image of *bla*TEM and *bla*CTX-M in *E. coli and K. pneumoniae* among patients and their family members.** L = Ladder with 1.2killo base DNA, Pc = Positive control, Nc = Negative control, bp = base pair, Lane: 1-17: Indicate specimen ID.

## Discussion

Antimicrobial resistance (AMR) is the most critical global public health threats [29]. Of particular concern are ESBL-PE and carbapenemase producing *E. coli* and *K. pneumoniae*, due to their resistance to broad range of beta-lactam antibiotics. These resistant bacteria commonly colonize the human gastrointestinal tract, often without causing symptoms [30]. However, asymptomatic individuals play a crucial role in the transmission of these resistant bacteria in the hospital settings where close interactions occur between patients and family members [31].

In the present study, intestinal colonization with ESBL-PE was observed among both patients and their families although it was significantly higher among the patients, 39.0% (39/100; 95% CI:29.5–49.3) as compared to their family members, 24.0% (24/100; 95% CI: 16.3–33.8 (P = 0.028). These findings suggest the burden of community acquired ESBL-PE colonization among both groups, which indicate the potential for resistant bacteria to be introduced into the hospital settings from the community. In addition, our findings highlight the need for antimicrobial stewardship implementation in the hospital and community settings.

The higher colonization rate of ESBL-PE among patients in our study was attributed to factors such as previous ICU admission and older age, both of which were found to be significantly associated with ESBL-PE colonization in our analysis. Our findings were close to the findings from a study in Burkina Faso, which reported ESBL-PE colonization rate of

**Fig 4. Gel electrophoresis image of *bla*~OXA-48~, *bla*~NDM~ and *bla*~KPC~ in *E. coli and K. pneumoniae* among patients at admission.** L = Ladder, PC = Positive control, NC = Negative control, bp = base pair. Lane:1-17 indicate the specimen ID.

42% among patients and 22% among healthy individuals [32]. However, the prevalence among patients in our study was much higher than the rate reported from a study in Germany, which found a 12.7% ESBL-PE colonization among patients at admission [33]. Similarly, ESBL-PE colonization rate in our observation among the family members was much higher than reported from previous studies in Europe, where prevalence of 9.1% [19] and 1.9% [34] were documented among household contacts. These discrepancies between our observation and those from resource-rich countries may indicate inappropriate use of antibiotics and low antimicrobial steward ship practice in our study setting in contrast to the developed nations that have well-established infection prevention and control strategies.

On the other hand, our finding on ESBL-PE colonization rate among patients was much lower than a 58.3% rate reported among hemodialysis patients from a study in Colombia; however, the rate among family members was comparable to household contacts (22.2%) from the latter study [35]. The discrepancy could be due to differences in study populations and study setting, as the Colombian study enrolled hemodialysis patients who might be exposed to invasive medical devices, whereas our study participants were patients with a variety of clinical conditions. Similarly, our findings were much more lower than ESBL-PE colonization rates reported from a study in Indonesian, where a prevalence of 88% and 76% were observed among hospitalized patients and their families, respectively [36]. The high prevalence from the Indonesian stud may be a result of widespread use of antimicrobials consumption without prescription in the community [37].

In this study, ESBL producing *E. coli* isolates were predominantly found more among patients than family members (P = 0.04). ESBL producing *K. pneumoniae* isolates were also common ly identified in patients than their family members. Frequent hospital exposure, previous history of antibiotic treatments and comorbidities might contribute for this variation. Moreover, these bacteria are commonly found in human gut, and widely distributed in the hospital and community settings [6]. These findings were parallel with the observation from a previous study that reported detection of ESBL producing *E.*

**Table 6. Association of factors with ESBL-PE colonization among patients at Tikur Anbessa Specialized Hospital in Addis Ababa, Ethiopia from February 2023 to October 2023 (n = 200).**

| Variables | | Patients (n = 100) | | | | | | | | |
|---|---|---|---|---|---|---|---|---|---|---|
| | | Total | ESBL-PE | | | | | | | |
| | | | Positive | Negative | COR | 95% CI | P-value | AOR | 95% CI | P-value |
| Sex | Female | 49 | 19 | 30 | 1.05 | 0.46-2.34 | 0.910 | | | |
| | Male | 51 | 20 | 31 | | | | | | |
| Age groups | 18-29 | 31 | 9 | 22 | | | | | | |
| | 30-41 | 32 | 12 | 20 | 1.47 | 0.51-4.07 | 0.470 | | | |
| | 42-53 | 27 | 10 | 17 | 1.44 | 0.49-4.21 | 0.510 | | | |
| | >53 | 16 | 8 | 2 | 9.78 | 1.71-55.80 | 0.010 | 8.91 | 1.52-52.3 | **0.020** |
| Previous antibiotic | Yes | 75 | 31 | 44 | 1.49 | 0.57-3.09 | 0.409 | | | |
| | No | 25 | 8 | 17 | | | | | | |
| Previous catheter use | Yes | 18 | 6 | 12 | 1.35 | 0.46-3.95 | 0.587 | | | |
| | No | 82 | 33 | 49 | | | | | | |
| Previous admission | Yes | 50 | 18 | 32 | 1.29 | 0.58-2.88 | 0.539 | | | |
| | No | 50 | 21 | 29 | | | | | | |
| Previous ICU admission | Yes | 25 | 20 | 5 | 11.78 | 3.88-35.75 | <0.001 | 12.4 | 3.98-38.6 | **<0.001** |
| | No | 75 | 19 | 56 | | | | | | |
| **Comorbidities (n = 100)** | | | | | | | | | | |
| HIV | Yes | 16 | 6 | 10 | 1.07 | 0.35-3.25 | 0.893 | | | |
| | No | 84 | 33 | 51 | | | | | | |
| Renal | Yes | 11 | 5 | 6 | 1.35 | 0.38-4.76 | 0.643 | | | |
| | No | 89 | 34 | 55 | | | | | | |
| Liver | Yes | 8 | 5 | 3 | 2.84 | 0.64-12.65 | 0.170 | | | |
| | No | 82 | 58 | 34 | | | | | | |
| Type 2 Diabetics | Yes | 6 | 4 | 2 | 3.37 | 0.58-19.36 | 0.173 | | | |
| | No | 94 | 59 | 35 | | | | | | |
| Cardiac problem | Yes | 12 | 4 | 8 | 1.32 | 0.36-4.72 | 0.669 | | | |
| | No | 88 | 35 | 53 | | | | | | |

**COR:** Crude odds ratio, **AOR:** Adjusted odds ratio, **CI:** Confidence interval.

coli and *K. pneumoniae* at 86.4%, the majority were from *E. coli* among hemodialysis patients, and the detection rate of 80.0% for ESBL producing *E. coli* among household contacts [35]. Other similar studies also reported high detection rate of 76% [19] and 78% [32] for ESBL producing *E. coli* among family members. However, both of the above studies reported lower detection rate of 19% [19] and 21% [32] for *K. pneumoniae* among family members.

Regarding CRE and CP-CRE detection rate at admission among patients in the present study, it was found to be 19.0% (19/100, 95% CI: 12.1–28.3), and 10.0% (10/100, 95% CI: 5.2–18.0), respectively. However, CRE colonization was not found in family members. These findings suggested that these patients were already carriers of CRE up on entering the hospital, although it is hard to trace the source of the bacteria given that their respective family members, which may represent the situation community, did not harbor them, which is in in line with our previous observation that only 0.8% of the study participants from the community were colonized with CRE [17]. The fact that such a substantial amount of patients are already colonized by CRE before admission may pose a risk for subsequent infections and transmission to other patients in the hospital environment [38]. Comparable prevalence of CRE at 17.2% and CP-CRE at 7% were reported from a previous study conducted among admitted patients in Ethiopia [16]. Similarly, a study in Egypt reported

**Table 7. Association of factors with ESBL-PE colonization among family members at Tikur Anbessa Specialized Hospital in Addis Ababa, Ethiopia from February 2023 to October 2023 (n = 200).**

| Variables | | Family members (n = 100) | | | | | |
|---|---|---|---|---|---|---|---|
| | | Total | ESBL-PE | | COR | 95% CI | P-value |
| | | | Positive | Negative | | | |
| Sex | Female | 48 | 12 | 36 | 1.11 | 0.44-2.78 | 0.822 |
| | Male | 52 | 12 | 40 | | | |
| Age group | 18-29 | 43 | 9 | 34 | | | |
| | 30-41 | 31 | 7 | 24 | 1.10 | 0.36-3.36 | 0.865 |
| | 42-53 | 16 | 4 | 12 | 1.25 | 0.32-4.85 | 0.738 |
| | >53 | 10 | 6 | 4 | 2.52 | 0.58-10.89 | 0.216 |
| Previous antibiotic use | Yes | 26 | 7 | 19 | 1.23 | 0.45-3.43 | 0.685 |
| | No | 74 | 17 | 57 | | | |
| Travel history | Yes | 8 | 3 | 5 | 1.29 | 0.23-7.12 | 0.770 |
| | No | 92 | 21 | 71 | | | |
| Admission history | Yes | 8 | 2 | 6 | 1.06 | 0.20-5.63 | 0.945 |
| | No | 92 | 22 | 70 | | | |

**COR:** Crude odds ratio, **CI:** Confidence interval.

**Table 8. Associated factors and CRE colonization among patients at Tikur Anbessa Specialized Hospital in Addis Ababa, Ethiopia from February 2023 to October 2023 (n = 200).**

| Variables | | Patients (n = 100) | | | | | |
|---|---|---|---|---|---|---|---|
| | | Total | CRE | | COR | 95% CI | P-value |
| | | | Positive | Negative | | | |
| Sex | Female | 49 | 9 | 40 | 0.97 | 0.35-2.63 | 0.951 |
| | Male | 51 | 10 | 41 | | | |
| Age group | 18-29 | 31 | 7 | 24 | 2.62 | 0.28-24.41 | 0.397 |
| | 30-41 | 32 | 6 | 26 | 2.07 | 0.22-19.67 | 0.524 |
| | 42-53 | 27 | 5 | 22 | 2.05 | 0.21-20.05 | 0.539 |
| | >53 | 10 | 1 | 9 | | | |
| Previous antibiotic use | Yes | 75 | 12 | 63 | 0.49 | 0.16-1.42 | 0.191 |
| | No | 25 | 7 | 18 | | | |
| Admission history | Yes | 50 | 9 | 41 | 0.87 | 0.32-2.38 | 0.798 |
| | No | 50 | 10 | 40 | | | |
| Previous ICU | Yes | 25 | 6 | 19 | 1.50 | 0.50-4.50 | 0.464 |
| | No | 75 | 13 | 62 | | | |
| Previous catheter use | Yes | 18 | 4 | 14 | 1.27 | 0.36-4.42 | 0.701 |
| | No | 82 | 15 | 67 | | | |
| HIV | Yes | 16 | 4 | 12 | 1.50 | 0.43-5.41 | 0.507 |
| | No | 84 | 15 | 69 | | | |
| Cardiac disease | Yes | 12 | 2 | 10 | 0.84 | 0.16-4.17 | 0.826 |
| | No | 88 | 71 | 17 | | | |
| Renal disease | Yes | 11 | 4 | 7 | 1.71 | 0.41-7.17 | 0.463 |
| | No | 89 | 16 | 73 | | | |
| Liver disease | Yes | 8 | 3 | 5 | 2.85 | 0.61-13.15 | 0.180 |
| | No | 92 | 16 | 76 | | | |

28% colonization rate of CRE among hospitalized patients within 48 hours of their admission, which was consistent with our findings even though there was a time difference in specimen collection [39]. In contrast, relatively lower colonization rate of CRE have been reported in countries where better antimicrobial stewardship and infection prevention policies are practiced such as in Italy (1.9%) [40], India (13.3%) [41] and China (10.8%) [42].

The most commonly identified CRE and CP-CRE bacterium was *K. pneumoniae*, with the proportion of 46.2% and 83.3%, respectively, while *E. coli* was identified in a small proportion of the participants comprising 16.8% of CRE and 38.5% of CP-CRE. The same predominance of *K. pneumoniae* as (15.4%) was reported from a previous study conducted in Ethiopia, which also reported *E. cloacae* (15.3%) and *E. coli* (12.4%) consistently followed behind *K. pneumoniae* [16]. However, this trend is reversed from studies in Egypt and China where *E. coli* was identified as a predominant CRE isolate (83.3%) followed by *K. pneumoniae* (17%) from Egypt [39], and 56.1% *E. coli* and 15.5% *K. pneumoniae*, from China [42]. These contrasting finding from various studies and countries may reflect differences in the distribution of CRE bacteria across different geographical locations.

With respect to level of phenotypic antimicrobial resistance detected in the current study, high level of resistance to third-generation cephalosporins were observed in ESBL producing isolates. Ceftriaxone was ineffective against 93.5% of ESBL producing *E. coli* from patients and 90.9% from family members. Likewise, higher resistance rate to ceftriaxone was found in ESBL producing *K. pneumoniae*, with 100% of isolates from both groups of study participants. The complete resistance level observed in ESBL producing *K. pneumoniae* is particularly concerning and reinforces its role as a high-risk pathogen in both community and hospital settings. These findings were isolated observations since previous study in Ethiopia also reported 100% resistance rate to ceftriaxone in ESBL producing *E. coli* and *K. pneumoniae* [43]. Similarly, high rates of ESBL producing *E. coli* and *K. pneumoniae* resistance to ceftriaxone were detected from Kenya at 69.6% and 91.3%, respectively [44].

Additionally, high resistance rate to non-β-lactam antibiotics such as ciprofloxacin was found in ESBL producing *E. coli* isolates from patients at the proportion of 58.1% compared to their family members, 40.0% (P > 0.05). A significant resistance rate of ESBL producing *E. coli* to nitrofurantoin was observed among from patients, at 54.8%, compared to 27.3% in isolates from their family member (P = 0.040). These higher resistance rates could be due to the selective pressure from previous antibiotic exposure, especially in admitted patients, and co-resistance mechanisms often found in association with ESBL-PE [44,45]. Furthermore, overuse and misuse of these antibiotics in both the community and hospital settings may facilitate the selection pressure for these resistant bacteria [46].

Globally, the most common ESBL encoding genes are $bla_{CTX-M}$, $bla_{TEM}$ and $bla_{SHV}$; $bla_{CTX-M}$ is a predominant gene in both the community and hospital settings, where it. has been increasingly reported from healthy individuals and highly associated with community acquired infections [47]. In the present study, $bla_{CTX-M}$ was the most frequently detected in ESBL producing *E. coli* isolates from patients at 74.2% compared to their family members at 68.2% (P > 0.05). These findings were consistent with a report from an Indonesian study that found $bla_{CTX-M}$ prevalence at 72.0% from patients and 52.0% from family members [36]; and from ESBL producing *E. coli* isolates in Chad at 40% and 49% among patients and healthy individuals, respectively [12]. A similar finding of 40.9% was reported in ESBL producing *E. coli* isolates from Colombia [35].

Another commonly detected ESBL encoding gene in the current study was $bla_{TEM}$, which was detected in 37.0% of ESBL producing *E. coli* isolates from patients and 40.0% of family members. Similarly, the $bla_{SHV}$ gene was detected in ESBL producing *E. coli* isolates from patients and family members, with the proportion of 32.2% and 22.7%, respectively. These findings were consistent with a previous study in Colombia that reported the prevalence of $bla_{TEM}$ in ESBL producing *E. coli* isolates from patients and households at 45.4% and 35%, respectively; and $bla_{SHV}$ at 31.7% among patients [35]. Comparable detection rates were reported from a study in Indonesia, where $bla_{TEM}$ was observed from 42.0% of patients and in 36.0% of family members, and $bla_{SHV}$ from 20.0% of patients and 18.0% of family members [36].

Moreover, $bla_{CTX-M}$ was detected in 100% ESBL producing *K. pneumoniae* isolates from family members, whereas only 87.0% of patients harbored ESBL producing *K. pneumoniae* isolates. Additionally, the same pattern

of $bla_{TEM}$ detection was observed in ESBL producing *K. pneumoniae* isolates, where 100% of the isolates from family members and 62.0% from patients harbored the gene. The higher proportion of $bla_{CTX-M}$ and $bla_{TEM}$ in family members could be due to the small number of *K. pneumoniae* isolates obtained from this study participants in contrast to the number of isolates obtained from patients. Similar to our study, $bla_{CTX-M}$ was identified at 60.0% and 27.0% of ESBL producing *K. pneumoniae* isolates among healthy individuals and hospitalized patients at hospital settings in Chad [12].

Another key finding in this study was that carbapenemase encoding genes, $bla_{NDM}$ and $bla_{OXA-48}$, were predominantly found in ESBL producing *K. pneumoniae* isolates from patients only, where 83.3% and 50.0%, respectively, of the isolates from patients harbored the genes. These genes were also detected from ESBL producing *E. coli* isolates obtained from patients only, although at lower rates of 30.7% and 23.1%, respectively. Our findings were inconsistent with the finding from a previous study, which reported 40% of $bla_{OXA-48}$, and 10% $bla_{NDM}$ among patients, while $bla_{OXA-48}$, were found in 14.3% of family members [48]. A study conducted in Egypt reported that 70% and 23.3% of *E. coli* isolates carried $bla_{OXA-48}$ and $bla_{NDM}$ respectively, which were different from our findings among admitted patients [39]. These inconsistent findings could be due to differences in geographical location, extensive antibiotic use and laboratory detection methods [49,50].

## Limitation

The study was conducted at a single hospital with small sample size, which may limit the generalizability of the findings to the general population. Future study may focus on multiple health facilities with large sample size to link the findings to the large population. The other limitation in this study is that it was unable to include molecular typing of isolates to assess transmission dynamics or genetic relatedness between strains of isolates from patient and those family members.

## Conclusion

This study revealed significant ESBL-PE colonization among both patients at admission and their respective family members, albeit the rate of detection from patient was significantly greater than that from family members. However, CRE and CP-CRE colonization was detected only among patients, where $bla_{NDM}$ and $bla_{OXA-48}$ were the predominantly detected genes. The most prevalent beta-lactamase gene in both patients and family members was $bla_{CTX-M}$. Additionally, Previous ICU admission and older age were significantly associated with ESBL-PE colonization among patients. This study underscores the need for fecal screening of ESBL-PE and CRE for patients at admission and their family members, which may limit subsequent infection and transmission of these resistant bacteria between patients, health care workers and family members.

## Supporting information

**S1 Table. Primer sequences used for amplification of β-lactamase genes.**
(DOCX)

**S2 Table. Primer sequences used for amplification of CP-CRE genes.**
(DOCX)

## Acknowledgments

We would like to extend our sincere appreciation to Addis Ababa University, Tikur Anbessa specialized hospital, Ethiopian Public Health Institute, Armauer Hansen Research Institute, and Arsho Diagnostic Laboratory for providing the essential materials for this research. Our gratitude also goes to the study participants and data collectors for their willingness to engage in this research endeavor.

## Author contributions

**Conceptualization:** Dessie Abera, Abel Abera Negash, Woldaregay Erku Abegaz.

**Data curation:** Dessie Abera, Surafel Fentaw, Eyob Beyene.

**Formal analysis:** Dessie Abera, Abel Abera Negash.

**Investigation:** Dessie Abera, Surafel Fentaw, Abel Abera Negash.

**Methodology:** Dessie Abera, Abel Abera Negash, Woldaregay Erku Abegaz.

**Supervision:** Adane Mihret, Abel Abera Negash, Woldaregay Erku Abegaz.

**Writing – original draft:** Dessie Abera.

**Writing – review & editing:** Adane Mihret, Abel Abera Negash, Woldaregay Erku Abegaz.

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
