## [Decision Letter · Decision Letter 0]

9 Dec 2025

β-lactamase producing Enterobacterales among adult patients and their family members at Tikur Anbessa Specialized Hospital, Addis Ababa, Ethiopia

PLOS One

Dear Dr. Abera,

Thank you for submitting your manuscript to PLOS ONE. After careful consideration, we feel that it has merit but does not fully meet PLOS ONE’s publication criteria as it currently stands. Therefore, we invite you to submit a revised version of the manuscript that addresses the points raised during the review process.

We look forward to receiving your revised manuscript.

Kind regards,

Reham Mokhtar ELTarabili

Academic Editor

PLOS One

**Journal Requirements:**

2. We note that your Data Availability Statement is currently as follows:

“All relevant data are within the manuscript and its supporting information files”

**Additional Editor Comments:**

Two reviewers assessed your manuscript. They provided a number of valid comments and suggestions that must be carefully addressed by the authors upon revision.

Reviewers' comments:

Reviewer's Responses to Questions

**Comments to the Author**

1. Is the manuscript technically sound, and do the data support the conclusions?

Reviewer #1: Yes

Reviewer #2: Yes

2. Has the statistical analysis been performed appropriately and rigorously?

Reviewer #1: Yes

Reviewer #2: Yes

3. Have the authors made all data underlying the findings in their manuscript fully available?

Reviewer #1: Yes

Reviewer #2: No

4. Is the manuscript presented in an intelligible fashion and written in standard English?

Reviewer #1: Yes

Reviewer #2: Yes

Reviewer #1: First I would like to thank PLOS One editors for inviting me to review this paper. I have comments for consideration during decision.

Title:

β-lactamase producing Enterobacterales among adult patients and their family members at Tikur Anbessa Specialized Hospital, Addis Ababa, Ethiopia

The title is interesting because it include traced study participants with index patients. And this could generate important evidence.

Abstract

Well written and address all expectation in this section. However, there is redundancy in the result and conclusion, please review.

Introduction

The introduction is good. However, the rational of the study need more clarification. There are similar study done in the community. Please indicate the gaps you want to fill in this regard. The justification look the same for ESBL-PE and CRE; in the title only indicated ESBL-PE, why not CRE.

Methods and Materials

In the method section, how you determine the sample size is not indicated. Somewhere in document sample size was 100, how you come up with this number? In the bacterial identification you mentioned using conventional method and VITEK 2 Compact system (bioMe´rieux, France). How you interprete the result is not explained.

In the PCR assay, you applied multiplex PCR assay. How you grouped the primers of the interest, based on what?

Results

Well presented

Reviewer #2: General comments

1. The paper has typographical errors. It should be corrected.

2. Clarification needed for sample size calculation.

3. Some genes classified as ESBL, namely blaOXA, blaVIM, and blaDIM, are actually carbapenemase genes, while blaAmpC belongs to the AmpC β-lactamase category and is distinct from ESBLs.

4. Tables showing phenotyping results along with chi-square, bivariate, and multivariate p-values should be included in the manuscript.

5. The study site, Tikur Anbessa Specialized Hospital, should be properly acknowledged.

**Do you want your identity to be public for this peer review?** For information about this choice, including consent withdrawal, please see our Privacy Policy

Reviewer #1: **Yes:** Tizazu Zenebe Zelelie

Reviewer #2:No

---

## [Author Response · Author response to Decision Letter 1]

18 Dec 2025

We appreciate the reviewers and the editor for their detailed review on this manuscript

---

## [Decision Letter · Decision Letter 1]

11 Jan 2026

Extended-spectrum β-Lactamase and Carbapenemase-producing Enterobacterales among adult patients and their family members at Tikur Anbessa Specialized Hospital, Addis Ababa, Ethiopia

PONE-D-25-40715R1

Dear Dr. Abera,

We’re pleased to inform you that your manuscript has been judged scientifically suitable for publication and will be formally accepted for publication once it meets all outstanding technical requirements.

Kind regards,

Reham Mokhtar ELTarabili

Academic Editor

PLOS One

Additional Editor Comments (optional):

Two reviewers accessed your manuscript and recommended acceptance. All concerns raised regarding the manuscript have been thoroughly addressed, and I recommend its acceptance for publication.

Reviewers' comments:

Reviewer's Responses to Questions

**Comments to the Author**

Reviewer #1: All comments have been addressed

Reviewer #2: All comments have been addressed

2. Is the manuscript technically sound, and do the data support the conclusions?

Reviewer #1: Yes

Reviewer #2: Yes

3. Has the statistical analysis been performed appropriately and rigorously?

Reviewer #1: Yes

Reviewer #2: Yes

4. Have the authors made all data underlying the findings in their manuscript fully available?

Reviewer #1: Yes

Reviewer #2: Yes

5. Is the manuscript presented in an intelligible fashion and written in standard English?

Reviewer #1: Yes

Reviewer #2: Yes

Reviewer #1: All my concerns have been well addressed. However, in the sample calculation, they use previus study done outside Ethiopia. Are they sure that there is no study in Ethiopia for it?

Reviewer #2: I would like to commend the author for their diligent efforts in revising the manuscript. All my comments have been thoroughly addressed, and the revisions have substantially improved the clarity, coherence, and overall quality of the work. Compared with the previous version, the manuscript is now significantly stronger and more refined. In its current form, the manuscript meets the required standards and can be considered suitable for publication.

**Do you want your identity to be public for this peer review?** For information about this choice, including consent withdrawal, please see our Privacy Policy

Reviewer #1: No

Reviewer #2: No

---

## [Editor Report · Acceptance letter]

PONE-D-25-40715R1

PLOS One

Dear Dr. Abera,

I'm pleased to inform you that your manuscript has been deemed suitable for publication in PLOS One. Congratulations! Your manuscript is now being handed over to our production team.

Kind regards,

on behalf of

Dr. Reham Mokhtar ELTarabili

Academic Editor

PLOS One